# GIE-Bench: Towards Grounded Evaluation for Text-Guided Image Editing

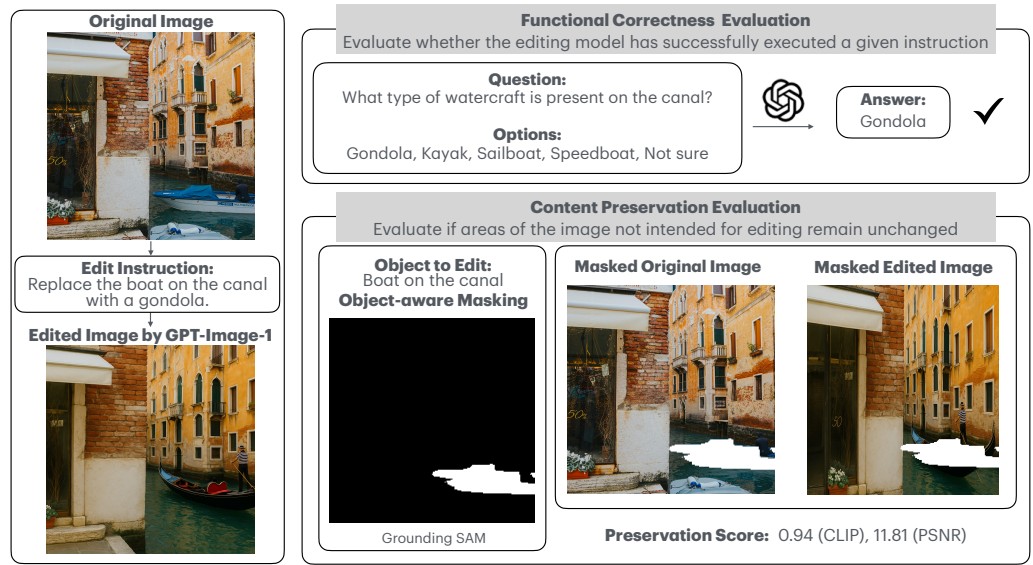

Figure 1: Overview of the proposed GIE-Bench pipeline for grounded and fine-grained evaluation of text-guided image editing models. GIE-Bench consists of two components: ($i$) functional correctness evaluation; and ($ii$) content preservation evaluation.

## Abstract

Editing images using natural language instructions has become a natural and expressive way to modify visual content; yet, evaluating the performance of such models remains challenging. Existing evaluation approaches often rely on image-text similarity metrics like CLIP, which lack precision. In this work, we introduce a new benchmark designed to evaluate text-guided image editing models in a more grounded manner, along two critical dimensions: ($i$) *functional correctness*, assessed via automatically generated multiple-choice questions that verify whether the intended change was successfully applied; and ($ii$) *image content preservation*, which ensures that non-targeted regions of the image remain visually consistent using an object-aware masking technique and preservation scoring. The benchmark includes over 1000 high-quality editing examples across 20 diverse content categories, each annotated with detailed editing instructions, evaluation questions, and spatial object masks. We note that our benchmark does not cover global editing tasks such as full style transfer, which remain important but are outside our current scope. We conduct a large-scale study comparing GPT-Image-1 (OpenAI, 2025), the latest flagship in the text-guided image editing space, against several state-of-the-art editing models, and validate our automatic metrics against human ratings. Results show that GPT-Image-1 leads in instruction-following accuracy, but often over-modifies irrelevant image regions, highlighting a key trade-off in the current model behavior. GIE-Bench provides a scalable, reproducible framework for advancing more accurate evaluation of text-guided image editing.

# 1 INTRODUCTION

Text-guided image editing has become a powerful and accessible way for users to modify images by simply describing the desired change in natural language, eliminating the need for complex tools or interfaces. Recent advances in diffusion models and multimodal large language models (MLLMs) have significantly improved the visual quality and instruction-following capability of these systems. Among them, GPT-Image-1 (OpenAI, 2025), a recently released model from OpenAI, has gained significant attention for its impressive performance in image editing. However, fine-grained evaluation of its editing performance remains limited.

Most prior works evaluate text-guided image editing performance using CLIP-based (Radford et al., 2021) similarity metrics or human preference scores. While useful, these approaches have key limitations. CLIP similarity offers only global alignment and does not reflect whether a model correctly executed the intended semantic change. Human evaluations, on the other hand, are costly, non-reproducible, and difficult to scale across thousands of examples.

To address this gap, we introduce **GIE-Bench**,[1] a new benchmark for evaluating text-guided image editing in a *grounded* manner, with precise and interpretable metrics. Our evaluation framework is twofold: ($i$) functional correctness evaluation, and ($ii$) content preservation evaluation.

First, we assess *functional correctness* using a VQA-style protocol. For each edit, we formulate a multiple-choice question grounded in the edited image, designed to verify whether the intended semantic change was applied. Unlike previous VQA-style evaluations that use binary yes/no questions (*e.g.*, in I2E-Bench (Ma et al., 2024)), our multiple-choice format includes two to five answer options, increasing the difficulty and reducing the chance of success by guessing. This enables automatic, objective, and more grounded assessment of instruction execution.

Second, we evaluate *image content preservation*, the model's ability to leave irrelevant areas of the image unchanged. Unlike prior work that applies global similarity metrics, we introduce an object-aware preservation score. Our benchmark provides masks for edited objects and computes similarity over the remaining area that is intended to be left untouched. This localized approach yields more sensitive and precise measurement of unintended changes. By explicitly separating the edited and preserved regions, GIE-Bench disentangles two competing dimensions of performance that prior benchmarks often confound.

To support robust evaluation, our benchmark covers over 800 diverse images across 20 categories (*e.g.*, human faces, animals, architecture, text, cartoon, art, food, and electronics), with over 1000 editing tasks spanning 9 functional edit types. Each sample includes natural language instructions, edit type metadata, object masks, a multiple-choice question for functional correctness evaluation, and human-annotated ground truth. This breadth of coverage ensures that evaluation reflects both fine-grained visual edits (e.g., color or attribute change) and more challenging semantic modifications (e.g., layout or scene change).

Finally, we benchmark and compare a wide array of state-of-the-art image editing models, including MGIE (Fu et al., 2024), OmniGen (Xiao et al., 2024), and the most recent GPT-Image-1 (OpenAI, 2025). Our results highlight GPT-Image-1's strong performance in functional correctness, while also revealing its tendency to over-edit irrelevant regions, as captured by our preservation metrics. This analysis demonstrates how GIE-Bench goes beyond surface-level comparisons and reveals subtle but important trade-offs in current systems.

Our contributions are summarized as follows.

- A high-quality image editing benchmark covering diverse domains and instruction types;
- A precise, fully automated evaluation protocol using VQA-style questions and object-aware masking to assess both functional correctness and preservation;
- A standardized empirical comparison of leading image editing models, with a detailed focus on GPT-Image-1's strengths and weaknesses.

---

[1] short for Grounded Image Editing Evaluation Benchmark. Benchmark data and evaluation code are released at https://anonymous.4open.science/r/GIE-Bench-127E.

| Benchmark | Instruction Eval | Preserv. Eval | Mask Usage in Eval | Edit Types Covered |
|---|---|---|---|---|
| I2E-Bench (Ma et al., 2024) | ✓(VQA) | ✗ | ✗ | Diverse |
| IE-Bench (Sun et al., 2025) | ✓(Human-rated) | ✓(MOS) | ✗ | Diverse |
| Edit-Bench (Wang et al., 2023) | ✓(Retrieval) | ✗ | ✗ | Caption-driven |
| DiffEdit (Couairon et al., 2022) | ✓(CLIP/FID curves) | ✓(LPIPS) | ✗ | Diverse |
| GIE-Bench (Ours) | ✓(VQA) | ✓ | ✓(SAM + DINO) | Diverse |

Table 1: Comparison of text-guided image editing evaluation benchmarks. Our benchmark uniquely evaluates both functional correctness and content preservation with precise mask-based protocols.

## 2 RELATED WORK

**Text-Guided Image Editing.** Early methods such as Prompt2Prompt (Hertz et al., 2022) and Imagic (Kawar et al., 2023) introduced ways to edit images with text prompts by modifying either the attention layers or the latent space of diffusion models. DiffEdit (Couairon et al., 2022) further refined image editing by detecting editable regions and generating masks. Text-guided image editing models have since emerged to simplify user interaction by directly following natural language instructions. InstructPix2Pix (Brooks et al., 2023) fine-tuned diffusion models on synthetic image editing pairs guided by instructions. More recent models Xiao et al. (2024); Geng et al. (2023); Fu et al. (2024; 2025); Le et al. (2024); Zhang et al. (2023); Huang et al. (2023); Li et al. (2024b); Lin et al. (2024); Meng et al. (2024); Han et al. (2024b); Li et al. (2024a); Guo & Lin (2023); Santos et al. (2024); Zhao et al. (2024) have advanced the semantic and compositional understanding of edits. InstructAny2Pix (Li et al., 2024b) uses multimodal inputs (*e.g.*, audio, image) for editing guidance. SmartEdit (Huang et al., 2023) focuses on complex scenes and incorporates MLLMs for better instruction comprehension. MGIE (Fu et al., 2024) learns to derive expressive instructions from user input and provides explicit guided visual editing. Most recently, OneDiffusion (Le et al., 2024) and UniVG (Fu et al., 2025) presents generalist diffusion models for unified image generation and editing.

**Benchmarks for Text-Guided Image Editing.** To address the need for image editing evaluation, several benchmarks Basu et al. (2023); Ma et al. (2024); Sun et al. (2025); Couairon et al. (2022) have been proposed. EditVal (Basu et al., 2023) assesses the fidelity of generated images across various edit types. It evaluates models based on their ability to perform specific attribute edits. I2E-Bench (Ma et al., 2024) emphasizes human perception alignment. IE-Bench (Sun et al., 2025) introduces a database containing diverse source images, editing prompts, and results from different editing methods, along with Mean Opinion Scores from human subjects. PIE-Bench (Ju et al., 2023) introduces a benchmark of 700 images with 10 editing types, annotated with masks and paired prompts, and primarily serves to evaluate inversion quality and diffusion-based editing fidelity. However, PIE-Bench does not explicitly assess instruction-following correctness beyond text-image similarity, and it does not disentangle functional correctness from unintended collateral changes. These benchmarks have advanced image editing evaluation, but often focus on either high-level text-image alignment or human perceptual scores without integrating functional correctness and content preservation in a unified framework. More related work on text-to-image generation benchmarks is provided in Appendix A.1.

*Evaluation of Content Preservation.* Evaluating content preservation is crucial to ensure that edits are confined to specified regions without unintended alterations. Traditional preservation metrics that compute CLIP (Radford et al., 2021) similarity over the entire image are flawed because they conflate intended edits with unintended changes, failing to isolate regions that should remain untouched.

*Evaluation of Functional Correctness via VQA.* Incorporating VQA into image editing evaluation offers a promising avenue for assessing functional correctness. By formulating the evaluation as a VQA task, models can be tested on their understanding of the edits in relation to the instructions. For example, TIFA (Hu et al., 2023) and Davidsonian Scene Graph (Cho et al., 2024) use VQA-based approaches to evaluate fine-grained alignment between text and images. I2E-Bench (Ma et al., 2024) also adopts a VQA-style evaluation by generating binary (yes/no) questions that query whether a

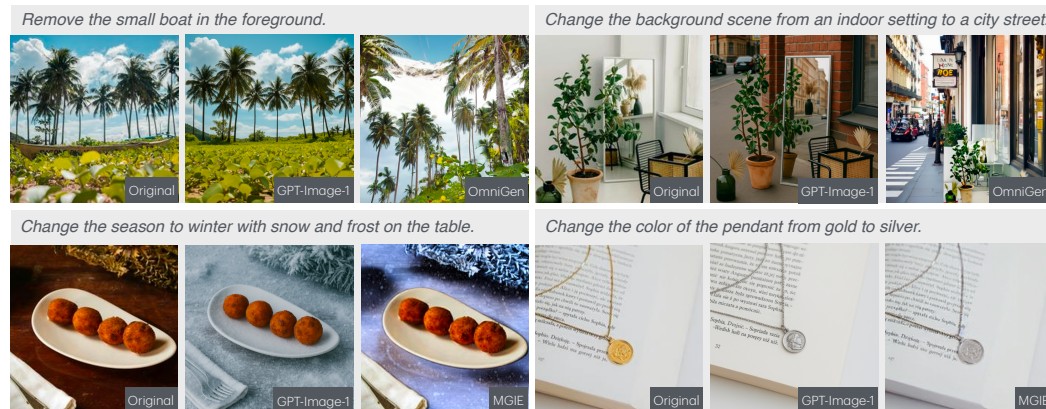

Figure 2: Example image editing instructions and edited results by GPT-Image-1 (OpenAI, 2025), OmniGen (Xiao et al., 2024), and MGIE (Fu et al., 2024).

particular edit was made. While effective, such binary formulations may oversimplify the evaluation and are easier to answer by chance. In contrast, our benchmark formulates each evaluation as a multiple-choice question with 2–5 carefully constructed options, requiring more nuanced reasoning and disambiguation.

**Our Contributions.** As shown in Table 1, most image editing benchmarks do not precisely evaluate whether the background is preserved while assessing instruction correctness. The use of segmentation masks to precisely isolate editable and non-editable regions is largely absent from these prior works.

GIE-Bench addresses these gaps by providing: ($i$) precise preservation evaluation by separating areas intended to edit from areas not intended to be edited by providing object masks, ($ii$) VQA-style multiple-choice questions to assess edits without requiring access to the original instruction. The dual-axis evaluation of GIE-Bench enables us to differentiate between models that make accurate edits but introduce unnecessary changes, and those that preserve image integrity but fail to satisfy the instruction. Our use of automated, mask-aware metrics makes our benchmark uniquely suited for large-scale, fine-grained evaluation of instruction following behavior in image editing models.

## 3 BENCHMARK CONSTRUCTION

### 3.1 DATASET COLLECTION

**Source of Images.** To ensure diversity in visual content and evaluate text-based image editing across a wide range of domains, we hand-selected 2000 images from Pexels, a repository of high-quality, royalty-free images. The images are of various resolution. They are evenly distributed across 20 distinct categories, with 100 images per category. These categories span a broad spectrum of real-world and synthetic visual content.

The selected categories are: ($i$) *Objects & Environments*: furniture, architecture, electronics, home appliance, city, nature, plant; ($ii$) *People & Clothing*: human, human face, cloth, accessories, jewellery; ($iii$) *Animals & Creatures*: animal, cartoon; ($iv$) *Food & Lifestyle*: food, musical instrument, art; ($v$) *Text & Symbols*: text, sign; ($vi$) *Vehicles*: transportation. This collection was curated to represent both structured and unstructured scenes, photographic and illustrated styles, and static and dynamic subjects.

**Generation of Editing Instructions.** Each image in our dataset is randomly paired with 3 editing tasks, from the 9 editing types: *Color Change*, *Add Object*, *Remove Object*, *Attribute Change*, *Object Replacement*, *Layout Modification*, *Scene/Background Change*, *Size Change*, and *Textual Edit* (only for images containing text). This variety is designed to comprehensively evaluate an image editing model's ability to handle both low-level appearance modifications and high-level semantic changes. We then prompt GPT-4o to generate edit instructions based on the image and edit type. We provide the detailed prompt in Appendix A.3. Figure 2 shows 4 examples of editing instructions and edited results.

## 3.2 GENERATION OF MULTIPLE-CHOICE QUESTIONS FOR FUNCTIONAL CORRECTNESS EVALUATION

To assess whether a model has correctly executed a given instruction, we introduce a VQA-style evaluation framework focused on functional correctness. For each instruction-image pair, GPT-4o generates a multiple-choice question that can only be answered correctly by inspecting the edited image, along with 2 to 5 answer choices including plausible distractors. The correct answer serves as ground-truth for automated evaluation. The quality of these questions is validated by testing them on original (unedited) images, yielding only 17% accuracy, confirming that successful answers require the correct edit. Prompt templates and examples are provided in Appendix A.3 and A.2.

## 3.3 OBJECT EXTRACTION AND MASK GENERATION FOR IMAGE CONTENT PRESERVATION EVALUATION

Traditional approaches to evaluating content preservation in text-based image editing often rely on computing similarity metrics across the entire edited image compared to the original. However, this strategy is fundamentally flawed. For example, if a model makes no edits at all, these metrics would yield a perfect similarity score, even though the editing instruction has been ignored. This makes it incorrect to assume that higher similarity scores necessarily indicate better performance. To address this, we propose a more precise and targeted evaluation method that focuses on the regions of the image that are not supposed to be edited. Our key idea is to isolate the editable region using segmentation masks, and then invert the mask to define the rest of the image: the portion that should remain unchanged. By comparing these preserved regions between the original and edited images, we obtain a much more faithful estimate of content preservation.

**Extracting and Segmenting Edit Targets.** To identify the region to be edited, we first extract the target object to be edited from each instruction using GPT-4, prompting it to return concise phrases (*e.g.,* "the green bus", "dog's tail"). These phrases are then passed to Grounded SAM (Liu et al., 2023) to produce a binary segmentation mask of the edit region. All masks are verified through human validation to ensure accuracy. This approach enables open-vocabulary grounding and supports precise, instruction-specific evaluation. The prompt used for GPT-4 is included in Appendix A.3.

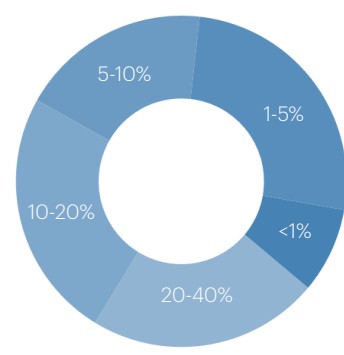

Object Mask Size Ratio

Figure 3: Distribution of object mask size ratios.

*Object Detection.* We use the object label predicted by GPT-4 (*e.g.,* "tree trunk") to perform open-vocabulary object detection using the GroundingDINO-Tiny model (Liu et al., 2023). GroundingDINO outputs bounding boxes along with confidence scores and associated class names. When multiple detections are returned for the same label, we select the one with the highest confidence score.

*Mask Generation.* Given the bounding box returned by GroundingDINO, we apply the Segment Anything Model (SAM-ViT-Base) (Kirillov et al., 2023) to produce a pixel-level segmentation mask for the object. This mask defines the editable region of the image and is stored as the object mask. We then compute and store the rest-of-image mask by inverting the object mask; this represents all parts of the image not intended for editing. The distribution of object mask size ratio is shown in Figure 3. Most object masks occupy less than 20% of the image area, indicating the prevalence of fine-grained edits.

This combination of using (*i*) GPT-4 for understanding open-domain natural language, (*ii*) GroundingDINO for open-vocabulary detection, and (*iii*) SAM for high-fidelity segmentation, allows us to robustly localize editable regions even when instructions involve abstract, uncommon, or compositional object references.

## 3.4 HUMAN FILTERING FOR HIGH-QUALITY ANNOTATIONS

To ensure high-quality evaluation data, we applied a human filtering step to our initially generated benchmark. We initially sampled 3 edit instructions for each image, resulting in a raw pool of 6,000 image-instruction pairs. However, we observed that some automatically generated object masks were inaccurate, and some edit instructions were ambiguous or invalid. To address these issues, we

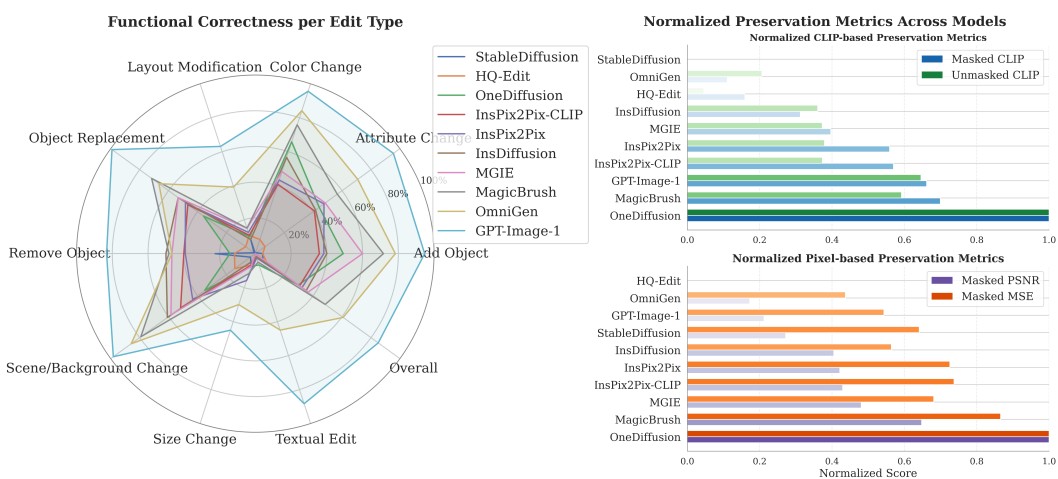

Figure 4: Left: Functional correctness per edit type. Right: Model preservation rankings based on masked MSE (inverted), PSNR, and CLIP scores. We empirically observe that model ranking evaluated by masked CLIP differs from model ranking evaluated by CLIP.

manually reviewed the entire dataset to only keep high-quality examples. The final curated benchmark consists of 1,080 high-quality image-instruction pairs with 856 unique images. We also made sure that these examples are carefully balanced across 9 edit types (120 edit instructions in each edit type) and 20 image categories.

## 3.5 EVALUATION METRICS

We report functional correctness and content preservation as separate metrics because combining them into a single score would obscure the trade-off between instruction following and image consistency.

**Functional Correctness.** To evaluate functional correctness, we adopt the VQA-style multiple-choice format described in Section 3.2. We use GPT-4o to answer each question, using only the edited image unless the question asks for a comparison between the original and edited images, and the question itself. We report accuracy of each edit type and overall accuracy across all benchmark entries.

**Image Content Preservation.** In addition to evaluating instruction fidelity, our benchmark assesses how well models preserve unedited content. To this end, we introduce a calibrated metric based on *masked, aligned mean squared error (MSE) difference*. Specifically, we compute the mean squared error between the original and edited images over unmasked regions, that is, regions that should remain unchanged, after applying geometric alignment.

To mitigate the impact of spatial shifts introduced during generation, we first align the edited image to the original using SIFT keypoints (Lowe, 2004), FLANN-based matching (Muja & Lowe, 2009), and affine transformation. This yields an partially aligned edited image $I'$ to the original image $I$. We then *invert and min-max normalize* the MSE values across all benchmark entries to produce a calibrated preservation score in the range $[0, 1]$, where higher values indicate better preservation.

In addition to masked MSE, we also compute masked SSIM, masked CLIP, and masked PSNR to evaluate both semantic and pixel-level preservation. Together, they provide a more comprehensive view of unintended changes introduced during editing and correlate strongly with human-annotated preservation scores (see Section 4.3).

## 4 EXPERIMENTS

### 4.1 MAIN RESULTS

**Functional Correctness.** Figure 4 (left) reports functional correctness evaluation results across various edit types, evaluated via multiple-choice QA with GPT-4o. Detailed numbers are reported in Table 2. Interestingly, GPT-image-1 maintains consistent high accuracy across all categories,

| Model | Add Object | Attribute Change | Color Change | Layout Modif. | Object Replace | Remove Object | Scene / Bkgd Change | Size Change | Textual Edit | Overall |
|---|---|---|---|---|---|---|---|---|---|---|
| StableDiffusion | 4.17 | 0.83 | 0.00 | 8.33 | 0.83 | 22.50 | 3.33 | 7.50 | 0.00 | 5.28 |
| HQ-Edit | 4.17 | 6.67 | 8.33 | 10.83 | 6.67 | 11.67 | 14.17 | 6.67 | 0.00 | 7.69 |
| OneDiffusion | 49.17 | 44.17 | 65.83 | 10.00 | 35.83 | 14.17 | 35.00 | 7.50 | 6.67 | 29.81 |
| InsPix2Pix-CLIP | 35.83 | 40.83 | 40.83 | 10.83 | 46.67 | 40.00 | 51.67 | 6.67 | 2.50 | 30.65 |
| InsPix2Pix | 38.33 | 47.50 | 43.33 | 12.50 | 48.33 | 39.17 | 43.33 | 15.83 | 2.50 | 32.31 |
| InsDiffusion | 40.00 | 41.67 | 56.67 | 8.33 | 53.33 | 50.00 | 60.83 | 4.17 | 1.67 | 35.19 |
| MGIE | 60.00 | 40.00 | 48.33 | 15.00 | 53.33 | 46.67 | 58.33 | 7.50 | 0.00 | 36.57 |
| MagicBrush | 70.83 | 57.50 | 75.00 | 13.33 | 72.50 | 48.33 | 82.50 | 9.17 | 5.83 | 48.33 |
| OmniGen | 78.33 | 70.83 | 84.17 | 39.17 | 66.67 | 46.67 | 85.83 | 30.00 | 45.00 | 60.74 |
| GPT-Image-1* | 92.98 | 94.59 | 92.04 | 64.08 | 97.32 | 73.45 | 96.36 | 45.95 | 83.33 | 82.42 |
| GPT-Image-1 | **94.74** | **95.50** | **95.58** | **63.11** | **99.11** | **83.19** | **98.18** | **45.05** | **88.33** | **85.00** |

Table 2: Functional correctness (multiple-choice accuracy) per edit type evaluated by GPT-4o. Values are raw percentages. * denotes judgment by Gemini-2-Flash Google (2024) as the second judge. Full Gemini-2-Flash results are in Appendix A.4.

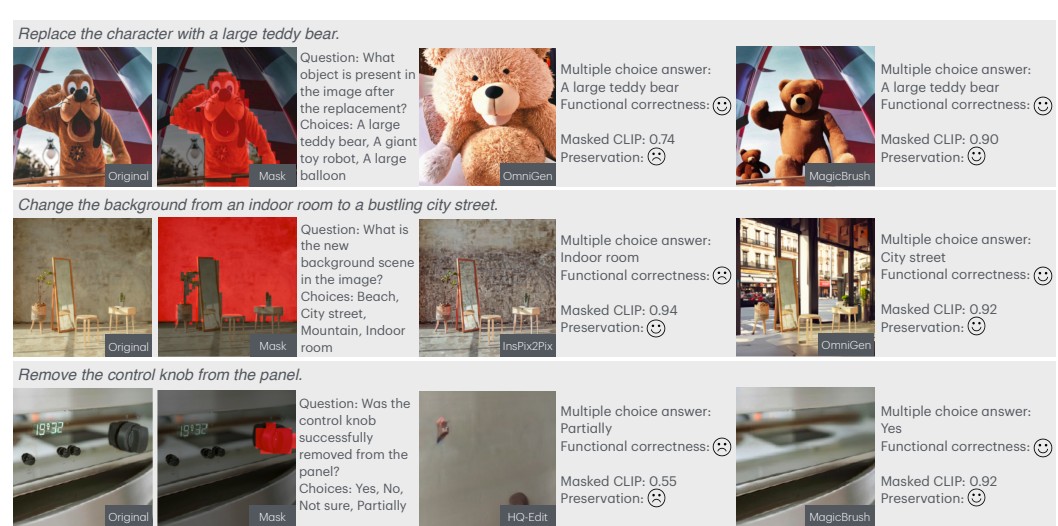

Figure 5: Examples showing three failure modes: functional failure, preservation failure, and combined failure, paired with correct editing.

while other models tend to vary more significantly depending on edit type, indicating a disparity in generalization capabilities. These findings highlight the importance of model architecture and instruction-following ability in achieving faithful functional edits.

In Figure 5, we show examples of both correct and incorrect edits, some failing in executing the instruction, and others failing to preserve areas not meant to be modified, emphasizing why a dual-axis evaluation is necessary.

**Image Content Preservation.**

Figure 4 (right) presents normalized preservation scores across models using both CLIP-based and pixel-based metrics. Detailed results are reported in Table 3. We observe that OneDiffusion and MagicBrush consistently achieve the highest preservation scores across all metrics. Interestingly, some models (*e.g.*, GPT-Image-1) score better on masked CLIP than pixel-based metrics, implying they preserve semantic structure better than low-level appearance. Depending on whether semantic preservation or exact pixel-level preservation is more important for the application, users can choose to evaluate preservation using CLIP-based or pixel-based metrics, respectively.

| Model | SSIM↑ | CLIP↑ | Unmasked CLIP↑ | PSNR↑ | MSE↓ |
|---|---|---|---|---|---|
| OneDiffusion | **0.9585** | **0.9805** | **0.9434** | **26.24** | **521.57** |
| MagicBrush | 0.8703 | 0.9489 | 0.8399 | 20.48 | 1529.45 |
| GPT-Image-1 | 0.5704 | 0.9485 | 0.8536 | 13.36 | 3952.34 |
| InsPix2Pix-CLIP | 0.7557 | 0.9375 | 0.7845 | 16.91 | 2498.30 |
| InsPix2Pix | 0.7537 | 0.9364 | 0.7859 | 16.78 | 2582.85 |
| MGIE | 0.7561 | 0.9189 | 0.7843 | 17.75 | 2917.03 |
| InsDiffusion | 0.7239 | 0.9099 | 0.7812 | 16.51 | 3799.92 |
| HQ-Edit | 0.4779 | 0.8949 | 0.7014 | 9.89 | 8035.06 |
| OmniGen | 0.4569 | 0.8943 | 0.7421 | 12.71 | 4752.33 |
| StableDiffusion | 0.6690 | 0.8790 | 0.6898 | 14.33 | 3219.74 |

Table 3: Preservation scores across five automatic metrics. All scores are computed over regions outside the object mask to evaluate unintended changes, except Unmasked CLIP which is widely used in previous work on image editing to evaluate preservation.

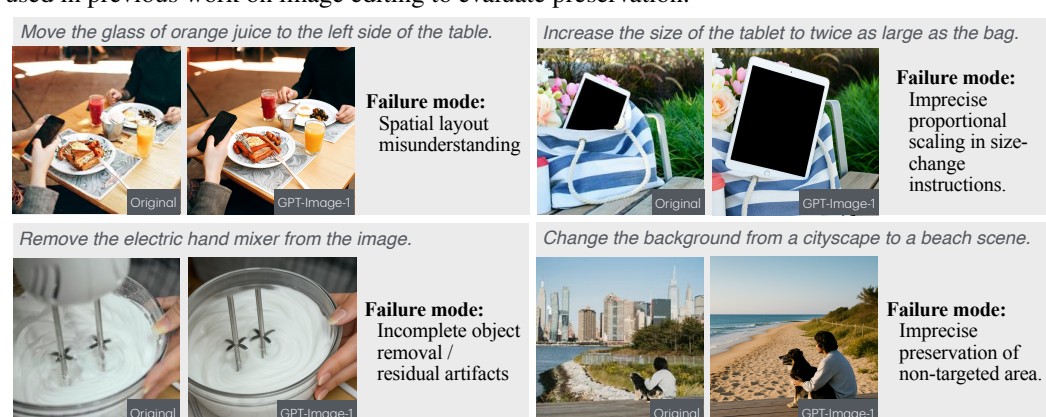

Figure 6: Examples of GPT-Image-1's failure modes.

## 4.2 HOW WELL DOES GPT-IMAGE-1 EDIT IMAGES?

GPT-Image-1 demonstrates strong overall performance across a wide range of text-guided editing instructions. It excels particularly in appearance-level edits, such as changing colors, modifying textures, adding or replacing objects, and editing text. For example, the model accurately changes the color of clothing items and signage, introduces new objects like potted plants or balloons with realistic rendering, and performs text edits with high spatial and semantic fidelity. These strengths highlight its powerful visual grounding and instruction-following ability.

However, GPT-Image-1 shows noticeable weaknesses in spatial and layout-related edits. For layout modifications (*e.g.*, "move the table to the left side" or "move the person to the right"), the model often interprets the direction loosely or inconsistently (example in Figure 6 top left). Similarly, in size-change instructions (*e.g.*, "make the vase twice as large as the lamp"), the modified objects sometimes appear disproportionate or fail to satisfy the relative sizing constraints (example in Figure 6 top right). These issues suggest that GPT-Image-1 struggles with precise spatial reasoning and proportional scaling.

In object removal tasks, GPT-Image-1 occasionally leaves residual artifacts or incompletely erases the object (example in Figure 6 bottom left), especially when the background is complex.

Another common failure is imprecision in preserving non-targeted regions. GPT-Image-1 frequently introduces unintended changes in areas unrelated to the instruction (example in Figure 6 bottom right), such as altering background textures, shifting lighting, or distorting peripheral objects. In contrast, models like MGIE tend to preserve these untouched regions more faithfully, leading to higher preservation scores and more stable image content.

Overall, while GPT-Image-1 is highly capable in executing core edits, it lacks fine-grained control over spatial relationships and content preservation, leaving room for improvement for tasks that demand high precision or minimal collateral changes.

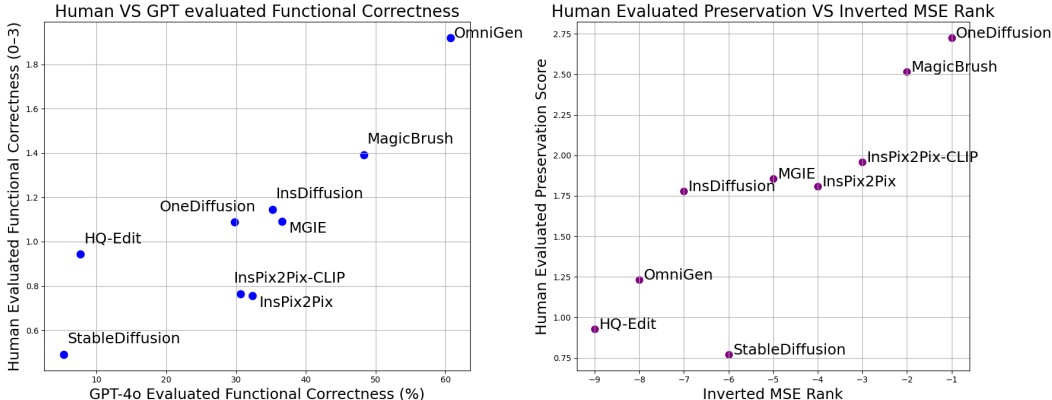

Figure 7: Left: Correlation between human and GPT-4o evaluated functional correctness scores. Right: Correlation between human-evaluated preservation and inverted calibrated MSE rank.

## 4.3 HUMAN STUDY

One of the central goals of our benchmark is to faithfully reflect human preferences when evaluating text-guided image editing. To assess this alignment, we conduct a human ranking study over a representative subset of 100 editing examples. Each example consists of an original image, an edit instruction, and the corresponding outputs generated by nine models[2]: StableDiffusion, HQ-Edit, OneDiffusion, InsPix2Pix-CLIP, InsPix2Pix, InsDiffusion, MGIE, MagicBrush, and OmniGen. Each image-edit pair is independently evaluated by four human annotators, who are instructed to provide two scores: one for instruction adherence and another for preservation of regions that are not supposed to be edited. Both scores used a 4-point scale, where 0 denotes complete failure, 1 indicates weak rejection, 2 represents weak acceptance, and 3 signals complete success. We then compute the average rank for each model across all 100 examples and annotators.

| Metric | Spearman |
|---|---|
| Masked SSIM | 0.881 |
| Masked CLIP | 0.952 |
| Masked PSNR | 0.905 |
| Masked Inverted MSE | 0.833 |

Table 4: Spearman correlation between human ratings on content preservation and automatic metrics.

Figure 7 (left) visualizes the relationship between GPT-4o evaluated functional correctness scores and human-annotated correctness, showing a clear positive trend. Table 4 reports the Spearman correlation coefficients between masked SSIM, masked CLIP, masked PSNR, and masked inverted MSE scores and human rankings (inverted so that higher values indicate better human preference alignment). Figure 7 (right) shows a strong correspondence between human preservation scores and inverted MSE rank, with models like OneDiffusion and MagicBrush scoring highly on both.

Together, these two metrics provide complementary views of model quality. Functional correctness captures whether the desired edit was made, while preservation evaluates whether it was done in a controlled and localized way. The strong alignment of both metrics with human preferences supports their joint use for comprehensive evaluation in text-guided image editing.

## 5 CONCLUSION

We introduce a benchmark for evaluating text-based image editing models along two critical axes: functional correctness and content preservation. Our evaluation combines VQA-style multiple-choice questions and object-aware preservation score to ensure edits are both instruction-faithful and minimally invasive. With over 1000 annotated examples across 20 diverse domains, the benchmark offers a scalable and interpretable framework. We also present a comparative analysis of leading editing models, revealing key strengths and weaknesses. We hope our benchmark drives progress toward developing text-based image editing systems that are not only visually compelling but also faithful to instructions and minimally invasive to unedited regions.

---

[2]GPT-Image-1 API was not available to us at the time thus not included.

## USAGE OF LLM IN PAPER WRITING

The authors used a LLM to help polish the text for grammar and style.

## LIMITATION

Our benchmark focuses on evaluating single-turn text-guided image edits where instructions are relatively atomic and localized. While this setting captures a wide range of edit types, it does not yet address more complex scenarios such as multi-step editing, interactive refinement, or long-form compositional instructions. In real-world applications, users often use sequential or iterative commands that require maintaining visual and semantic coherence over time. These scenarios pose additional challenges for instruction interpretation, memory, and image consistency, demanding more advanced model capabilities and evaluation methods. In future work, we plan to extend text-guided image-editing evaluation to these more dynamic settings.

## ETHICS STATEMENT

We constructed GIE-Bench using only royalty-free images, ensuring no copyrighted or personal data was included. Human annotators judged visual quality of edits without providing personal information and were compensated fairly. Our benchmark is intended for research use, aiming to advance reliable and safe image editing evaluation while mitigating risks of misuse.

## REPRODUCIBILITY STATEMENT

We release the full dataset, object masks, and evaluation scripts at https://anonymous.4open.science/r/GIE-Bench-127E to support reproducibility.

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

# A APPENDIX

## A.1 RELATED WORK ON BENCHMARKS FOR TEXT-TO-IMAGE GENERATION

Several benchmarks Saharia et al. (2022); Yu et al. (2022); Huang et al. (2023); Kirstain et al. (2023); Han et al. (2024a); Li et al. (2023); Wang et al. (2025); Hu et al. (2023); Cho et al. (2024); Ghosh et al. (2023) have been proposed to evaluate text-to-image generation models from different aspects. DrawBench (Saharia et al., 2022) and PartiPrompt (Yu et al., 2022) assess basic compositional abilities like attributes and spatial relationships. T2I-CompBench (Huang et al., 2023) introduces more complex reasoning skills such as comparison and logic. Pick-a-Pic (Kirstain et al., 2023) and EvalMuse-40K (Han et al., 2024a) collect large-scale human preferences for ranking outputs. Meanwhile, AGIQA-3K (Li et al., 2023) and EvalMi-50K (Wang et al., 2025) incorporate detailed perceptual and correspondence annotations, providing fine-grained and scalable evaluation of both image quality and instruction adherence. GenEval (Ghosh et al., 2023) offers an object-centric evaluation framework that assesses compositional properties. TIFA (Hu et al., 2023) and Davidsonian Scene Graph (Cho et al., 2024) adopt VQA-based methods to assess fine-grained text-image alignment.

## A.2 EXAMPLES FOR FUNCTIONAL CORRECTNESS EVALUATION

Table 5 provides representative examples of the nine edit types included in GIE-Bench, covering both low-level appearance changes and high-level semantic modifications. For each edit type, we show the input image content, the natural language instruction provided to the model, the corresponding multiple-choice question used for functional correctness assessment, and the expected answer.

| Edit Type | Image Content | Example Instruction | Evaluation Question and Options | Expected Answer |
|---|---|---|---|---|
| Color Change | A black coffee mug on a wooden table | Change the color of the mug to red. | What color is the mug? ['Black', 'Green', 'Red', 'Blue'] | Red |
| Add Object | A child standing in a grassy field | Add a yellow kite in the sky above the child. | What is flying in the sky above the child? ['A red balloon', 'Nothing', 'A yellow kite', 'A drone'] | A yellow kite |
| Remove Object | A dog and a ball on a lawn | Remove the ball from the lawn. | What objects are visible on the lawn? ['A ball', 'Nothing', 'A dog', 'A cat'] | A dog |
| Scene Change | A tree with green leaves in a sunny park | Change the season to winter. | What season is shown in the image? ['Spring', 'Autumn', 'Winter', 'Summer'] | Winter |
| Background Change | A man standing in front of a beach | Change the background to a snowy mountain. | What is in the background? ['Beach', 'Forest', 'Snowy mountain', 'City street'] | Snowy mountain |
| Attribute Change | A woman with a neutral expression | Make the woman smile. | Is the woman smiling? ['No', 'Not clear', 'Yes', 'Maybe'] | Yes |
| Object Replacement | A table with an apple on it | Replace the apple with a banana. | What fruit is on the table? ['Apple', 'Pear', 'Banana', 'Orange'] | Banana |
| Layout Modification | A dining table with a vase in the center | Move the vase to the left side of the table. | Where is the vase located on the table? ['Right side', 'On the floor', 'Left side', 'Center'] | Left side |
| Size Change | A single red apple on a white plate | Make the apple significantly larger to 2 times the size of the plate. | How big is the apple relative to the plate? ['Smaller than the plate', 'Slightly larger than the plate', 'Three times larger than the plate', 'Twice the size'] | Twice the size |
| Textual Edit | A notebook with a poem written on it | Change the first line of the poem to 'In love we believe' | What is the first line of the poem? ['In love we believe', 'In peace we believe', 'To be or not to be ', 'Not sure'] | In love we believe |

Table 5: Examples of edit instructions, evaluation questions and options, and expected answer of 9 edit types (Scene Change and Background Change are in the same category).

## A.3 PROMPT TEMPLATES TO GENERATE BENCHMARK

The prompts we used to generate editing instructions using GPT-4o are shown in Figure 8. The prompt we used to summarize object to edit based on the editing instruction is shown in Figure 9.

## A.4 ROBUSTNESS OF GPT-4O-BASED EVALUATION

To evaluate the stability of using GPT-4o for automatic functional correctness assessment in our VQA-style setup, we conducted three independent runs of the evaluation process. For each run, GPT-4o was provided with the same sets of multiple-choice questions and corresponding edited images for each model. We set the decoding temperature to 0 in all runs to ensure deterministic outputs.

**"Color Change"**: "Based on this image, identify an object with a clear color. Write an instruction to change its color. Then write a question to verify the new color and provide the correct answer and distractors. Format: {\"edit_instruction\":..., \"evaluation_question\":..., \"expected_answer\":..., \"multiple_choice_options\":..., \"edit_type\":\"Color Change\"}"

**"Add Object":** "Based on this image, describe a new object to be added. Write an instruction for the addition, a verification question, and 3 distractors plus the correct answer. Format: {\"edit_instruction\":..., \"evaluation_question\":..., \"expected_answer\":..., \"multiple_choice_options\":..., \"edit_type\":\"Add Object\"}"

**"Remove Object"**: "From the image, pick one clearly visible object to remove. Write the edit instruction, a question to confirm its removal, and distractors plus the correct answer. Format: {\"edit_instruction\":..., \"evaluation_question\":..., \"expected_answer\":..., \"multiple_choice_options\":..., \"edit_type\":\"Remove Object\"}"

**"Scene Change":** "Describe how to change the scene setting (e.g. season, weather, time of day). Write a corresponding instruction, question, and multiple-choice options. Format: {\"edit_instruction\":..., \"evaluation_question\":..., \"expected_answer\":..., \"multiple_choice_options\":..., \"edit_type\":\"Scene Change\"}"

**"Attribute Change"**: "Identify a modifiable attribute of an object or person (e.g., facial expression, texture). Write the instruction to change it, a VQA-style question, and answer options. Format: {\"edit_instruction\":..., \"evaluation_question\":..., \"expected_answer\":..., \"multiple_choice_options\":..., \"edit_type\":\"Attribute Change\"}"

**"Object Replacement"**: "Select an object in the image and write an instruction to replace it with another. Then write a question to verify what is present after the replacement, with distractors. Format: {\"edit_instruction\":..., \"evaluation_question\":..., \"expected_answer\":..., \"multiple_choice_options\":..., \"edit_type\":\"Object Replacement\"}"

**"Layout Modification"**: "Move or reposition an object within the image (e.g., move the vase to the left side). Write the instruction, evaluation question, and answer set. Format: {\"edit_instruction\":..., \"evaluation_question\":..., \"expected_answer\":..., \"multiple_choice_options\":..., \"edit_type\":\"Layout Modification\"}"

**"Background Change"**: "Alter the background scene in the image (e.g., from beach to city street). Write the instruction and a VQA-style question to confirm the new background. Format: {\"edit_instruction\":..., \"evaluation_question\":..., \"expected_answer\":..., \"multiple_choice_options\":..., \"edit_type\":\"Background Change\"}"

**"Size Change"**: "Select an object and significantly change its size (larger or smaller). Write an instruction and a question to verify the new size relative to nearby objects. Format: {\"edit_instruction\":..., \"evaluation_question\":..., \"expected_answer\":..., \"multiple_choice_options\":..., \"edit_type\":\"Size Change\"}"

**"Textual Change"**: "Select text in the image and write an instruction and a question to verify the edit made to the text (font, color, content). Format: {\"edit_instruction\":..., \"evaluation_question\":..., \"expected_answer\":..., \"multiple_choice_options\":..., \"edit_type\":\"Textual Change\"}"

Figure 8: Prompts used to generate editing instructions, multiple-choice questions, and true answers.

Given this image editing instruction: {instruction}. Please extract and describe the object or objects that need to be edited in the original image. Do not include objects to appear in the edited image. For example, 'red apple', or 'giraffe'. If the change is applied to the entire image like a lighting change, then say 'entire image'. Be concise.

Figure 9: Prompt used to summarize object to edit in the editing instructions.

| Model | GPT-4o Run 1 | GPT-4o Run 2 | GPT-4o Run 3 |
|---|---|---|---|
| StableDiffusion | 5.19 | 5.46 | 5.28 |
| HQ-Edit | 7.78 | 8.06 | 7.69 |
| InsDiffusion | 35.46 | 35.37 | 35.19 |
| OmniGen | 61.02 | 61.48 | 60.74 |

Table 6: Functional correctness accuracy (%) across three evaluation runs using GPT-4o with temperature set to 0.

The results show strong consistency across runs, with negligible fluctuations (typically under $\pm 0.3$ percentage points). The per-model scores are stable. This confirms that GPT-4o, when used deterministically, provides a robust evaluation, which strengthens the benchmark's reproducibility and fairness in model comparison.

| Model | Add Object | Attribute Change | Color Change | Layout Modif. | Object Replace | Remove Object | Scene / Bkgd Change | Size Change | Textual Edit | Overall |
|---|---|---|---|---|---|---|---|---|---|---|
| StableDiffusion | 6.67% | 5.83% | 4.17% | 13.33% | 5.00% | 21.67% | 7.50% | 15.83% | 3.33% | 9.26% |
| HQ-Edit | 11.67% | 10.00% | 16.67% | 11.67% | 10.00% | 15.00% | 17.50% | 17.50% | 3.33% | 12.59% |
| OneDiffusion | 44.17% | 43.33% | 64.17% | 10.83% | 35.83% | 11.67% | 37.50% | 22.50% | 9.17% | 31.02% |
| InsPix2Pix-CLIP | 38.33% | 42.50% | 39.17% | 15.83% | 48.33% | 30.00% | 47.50% | 20.83% | 7.50% | 32.22% |
| InsPix2Pix | 42.50% | 47.50% | 44.17% | 19.17% | 55.83% | 30.83% | 45.00% | 17.50% | 8.33% | 34.54% |
| MGIE | 49.17% | 36.67% | 50.00% | 19.17% | 50.83% | 40.00% | 61.67% | 22.50% | 2.50% | 36.94% |
| InsDiffusion | 48.33% | 45.00% | 67.50% | 15.83% | 59.17% | 44.17% | 67.50% | 13.33% | 2.50% | 40.37% |
| MagicBrush | 67.50% | 57.50% | 76.67% | 15.83% | 70.00% | 39.17% | 77.50% | 19.17% | 7.50% | 47.87% |
| OmniGen | 78.33% | 75.00% | 84.17% | 49.17% | 70.83% | 38.33% | 80.83% | 35.00% | 50.83% | 62.50% |
| GPT-Image-1 | 92.98% | 94.59% | 92.04% | 65.05% | 97.32% | 73.45% | 96.36% | 46.85% | 84.17% | 82.72% |

Table 7: Functional correctness (multiple-choice accuracy) per edit type as judged by Gemini-2-flash.

To assess the robustness of our functional correctness evaluation, we employed Gemini-2-Flash Google (2024) as a secondary judge alongside GPT-4o. For each edited image and its associated multiple-choice question, we recorded the answer selected by Gemini-2-Flash using the same VQA-style evaluation protocol. As shown in Table 3, the scores from Gemini closely track those of GPT-4o, confirming the reliability of our automatic evaluation framework across different models.

## A.5 HUMAN STUDY

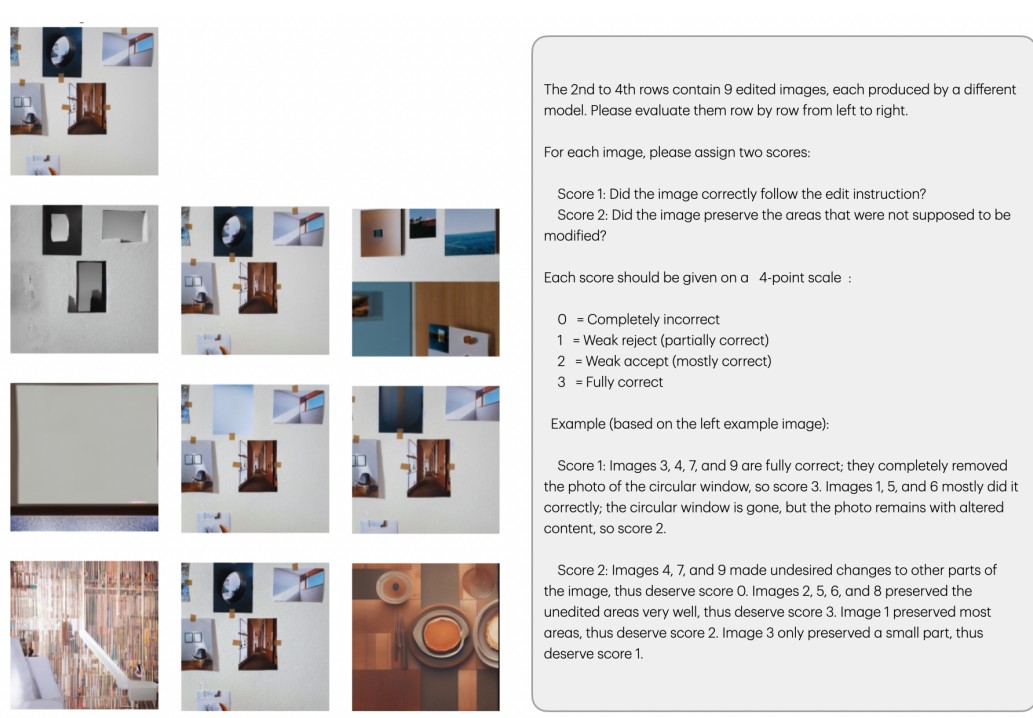

Figure 10: The annotation guidance we provided to human annotators on how to assign scores.

To validate the alignment of our automatic evaluation metrics with human perception, we conducted a human study over a randomly chosen subset of 100 edited examples from GIE-Bench.

**Annotation Setup.** For each example, participants were presented with: ($i$) one original image (top row), ($ii$) a set of nine edited images, each generated by a different image editing model (arranged in rows below), ($iii$) the editing instruction (*e.g.*, "Remove the photo of the circular window."), and the edit type (*e.g.*, Remove Object) displayed at the bottom.

The annotation guidance we provided on how to assign scores is shown in Figure 10. The human annotation task in this study involved scoring visual outputs of image editing models and did not collect any personal or sensitive information from participants. Hourly wage is 25 USD.

