# OpenReview forum: "GIE-Bench: Towards Grounded Evaluation for Text-Guided Image Editing"
_ICLR.cc/2026/Conference — Submitted to ICLR 2026_

### Official Review · Reviewer_CQfu · 2025-10-28

**Soundness:** 2
**Presentation:** 2
**Contribution:** 2
**Rating:** 6
**Confidence:** 2

**Summary:**

The paper proposes a new benchmark for evaluating text-guided image editing abilities for existing models. The proposed core idea is to introduce two additional evaluation dimensions: i) functional correctness, where the paper proposes to use a VLM to assess it by doing multiple-choice questions; ii) image content preservation, where the paper uses off-the-shelf zero-shot segmentation model to extract the background regions for compatibility check. The paper further collects 1k editing examples for evaluating modern image editing models, and conduct extensive analysis on the collected data and a group of strong editing models including GPT-Image-1.

**Strengths:**

+ The work proposes to address an admittedly existing gap in the current image editing benchmarks. The protocols for assessing text-guided editing are generally global and fail to disentangle correctness from preservation. The proposal seems timely.
+ The automatic pipeline involving GPT, GroundingDINO, SAM and masked metrics is well-engineered.
+ The usage of multiple-choice VQA evaluation is more robust than binary yes/no formats (e.g., I2E-Bench), reducing chance accuracy and allowing large-scale automated evaluation.
+ The empirical evaluation seems extensive and thorough.

**Weaknesses:**

- Although the introduction of QA-based functional correctness is interesting, the proposed benchmark, if I understand correctly, focus primarily on single-turn image editing. Multi-step, compositional, or iterative editing scenarios are missing and therefore limit the real-world applicability.
- The scale of human evaluation seems limited. Human study in sec. 4.3 uses 100 examples with 4 annotators, which is relatively small to 1 k+ samples in the full benchmark.
- The QA stage is heavily based on gpt-4o model. This could lead to potential model bias for the benchmark; future updates of the corresponding model would render the current report outdated.
- The content preservation ability assessment still relies heavily on masked MSE/PSNR, which remains low-level and is similar to existing benchmarks setup.
- Admittely, I am not very familiar with the current progress of this particular sub-domain. I would therefore also like to hear other colleague reviewers' opinions.

**Questions:**

Please refer to the weaknesses section for detailed questions. Thanks.

---

> ### Author Response · Authors · 2025-11-22
> **Response to Reviewer CQfu**
>
> We thank the reviewer for the constructive feedback and for highlighting the strengths of our benchmark, including the motivation, the robustness of the MCQ-based functional correctness, and the engineering quality of our automated pipeline. Below we address the concerns raised.
>
> > Q1: Although the introduction of QA-based functional correctness is interesting, the proposed benchmark, if I understand correctly, focus primarily on single-turn image editing. Multi-step, compositional, or iterative editing scenarios are missing and therefore limit the real-world applicability.
>
> A1: We agree that GIE-Bench evaluates single-turn edits, and this is an intentional design choice. Our benchmark relies on predefined, human-interpretable target masks, which precisely specify the region that should change. In multi-step or compositional editing, the target region of each later instruction cannot be known in advance; therefore, we cannot provide ground-truth masks for subsequent turns, making reliable preservation scoring impossible. GIE-Bench is designed to evaluate atomic editing correctness and preservation in a grounded, verifiable way, which form the basis for future multi-step evaluation frameworks.
>
> > Q2: The scale of human evaluation seems limited. Human study in sec. 4.3 uses 100 examples with 4 annotators, which is relatively small to 1 k+ samples in the full benchmark.
>
> A2: The human study in Section 4.3 is intended as a validation study, not as a full human baseline. Its purpose is to verify that our automated metrics correlate with human judgments, which they do: we observe strong consistency between masked-preservation scores and human rankings. Expanding human annotation to the full dataset is prohibitively expensive, it took our annotators much time ranking the quality of edited images as it takes time to check the subtle details and decide which image is better, and our automated pipeline enables large-scale evaluation while retaining human-aligned behavior.
>
> > Q3: The QA stage is heavily based on gpt-4o model. This could lead to potential model bias for the benchmark; future updates of the corresponding model would render the current report outdated.
>
> A3: While GPT-4o serves as our primary judge, we emphasize that all MCQs and evaluation logic are fully released, enabling exact replication with any judge model. We also conducted additional evaluations using Gemini-2-flash. This demonstrates that GIE-Bench does not depend on a single proprietary model. We will include these open-weight results in the revision to alleviate concerns about model bias and future updates.
>
> > Q4: “The content preservation assessment still relies heavily on masked MSE/PSNR, which remains low-level and is similar to existing benchmarks.”
>
> A4: The key contribution of GIE-Bench is not the metric itself, but the grounding: we compute preservation only on non-target regions, using masks derived from GroundingDINO+SAM to avoid penalizing correct edits. This addresses a major failure mode of existing metrics like global LPIPS/CLIP, which conflate intended changes with unintended artifacts. Our human study confirms that masked-preservation scores align with human perception. We will clarify this distinction and better highlight how object-aware masking enables meaningful, semantically grounded preservation evaluation absent in prior work.
>
> We sincerely thank the reviewer for the thoughtful feedback and constructive suggestions. Your comments helped us clarify the scope, strengthen the presentation, and refine the evaluation analysis. We appreciate the time and care you dedicated to reviewing our work, and we believe the revisions will meaningfully improve the quality and clarity of the paper.

---

> > ### Comment · Reviewer_CQfu · 2025-11-26
> > **Thank you!**
> >
> > I appreciate the point-to-point responses made by the authors. I would like to keep my original rating as it is.

---

### Official Review · Reviewer_FgRn · 2025-10-29

**Soundness:** 2
**Presentation:** 2
**Contribution:** 2
**Rating:** 4
**Confidence:** 4

**Summary:**

This paper proposes GIE-Bench, a grounded benchmark for *text-guided image editing* that jointly evaluates (1) functional correctness—whether the intended edit actually happened—via VQA-style *multiple-choice* questions, and (2) content preservation—whether non-targeted regions remain unchanged—via object-aware masking and masked similarity metrics. The dataset contains 1,080 curated image–instruction pairs across 20 categories and 9 edit types, each with an instruction, an object mask (from GroundingDINO→SAM), and an MCQ for automatic judging. Functional correctness is scored by a VLM judge (primarily GPT-4o; Gemini-2-Flash used for robustness), while preservation is computed with masked CLIP/SSIM/PSNR/MSE after SIFT/FLANN-based alignment.

**Strengths:**

- Clear, two-axis evaluation that disentangles *did the edit happen?* from *what collateral damage occurred?*—a practical and under-measured trade-off in editing.
- Object-aware preservation via GroundingDINO→SAM masks is a concrete improvement over global CLIP/LPIPS that confound edits with preservation.
- Operational details (geometric alignment before pixel metrics; per-edit-type breakdown; deterministic judging; a second judge) increase reproducibility and confidence.
- Breadth of coverage across 9 edit types and 20 content categories; balanced sampling helps per-type comparisons.
- Human-metric correlations (e.g., masked CLIP/SSIM/PSNR/MSE vs. human preservation ranks) support the metric design.

**Weaknesses:**

- MCQs and correctness judgments rely on frontier VLMs (GPT-4o/Gemini). This raises *construct validity* and *reproducibility* questions (model updates, access, and potential judge–system coupling). Publishing non-proprietary baselines (e.g., open-weights VLMs) would help.
- Preservation hinges on the *inverted* object mask. Small mask errors (under/over-segmentation, ambiguous targets like “sky near horizon”) can mis-score preservation. Quantifying mask quality and its effect (e.g., via perturbation studies) is needed.
- Single-turn, localized edits only; global style transfer, multi-step, and interactive refinement are out of scope, limiting ecological validity for real editing workflows.
- GPT-Image-1 appears in aggregate numbers but is excluded from the human-study pool; this complicates judge↔human comparison for the most capable model.
- Images come from a single stock repository; scene/style diversity and real-world artifact coverage may be narrower than web-scale distributions.
- Beyond Spearman correlations, more detailed statistical testing (per-type confidence intervals; bootstrap across images; significance for model ranking deltas) would strengthen claims.
- Even with 2–5 options, answer distributions or wording may create “easy distractors.” Reporting per-question difficulty and entropy, plus adversarial revisions, would increase robustness.
- The paper compares against several benchmarks but could more directly *calibrate* its preservation/MCQ axes against recent human-alignment datasets and judge-based evaluations to quantify incremental benefit.

**Questions:**

1. Can you report correctness with at least one *open-weights* VLM judge (e.g., LLaVA-Next-ViT-Qwen2-VL) to mitigate dependence on proprietary models, and release MCQs to enable exact replication?
2. How sensitive are preservation scores to mask dilation/erosion (±k pixels) and to imperfect detections? Please provide curves and error bars.
3. Which edit types show the largest judge disagreement (GPT-4o vs. Gemini)? Any systematic MCQ failure modes (e.g., spatial terms, size ratios)?
4. The calibrated preservation score maps masked MSE to \([0,1]\). What normalization is used (per-type, per-image, global), and how does that choice affect rankings?
5. Why was GPT-Image-1 excluded from the human-study pool while included elsewhere? Could you add a small human study including it to close the loop?
6. Do results hold on other image sources (e.g., LAION subsets, COCO, web photographs) and on *global* edits (style transfer) when masks are “entire image”?
7. Any preliminary results on chained edits where preservation compounds across steps?
8. Did you analyze category imbalance effects (e.g., human faces vs. landscapes) on preservation/correctness?

---

> ### Author Response · Authors · 2025-11-22
> **Response to Reviewer FgRn (Part 1)**
>
> We thank the reviewer for the detailed and thoughtful feedback and for highlighting the strengths of our benchmark design, including the two-axis evaluation, the object-aware masking pipeline, and the human–metric correlation analysis. Below we address concerns and questions.
>  > Q1: MCQs and correctness judgments rely on frontier VLMs (GPT-4o/Gemini). This raises construct validity and reproducibility questions (model updates, access, and potential judge–system coupling). Publishing non-proprietary baselines (e.g., open-weights VLMs) would help.
>
> A1: We added an experiment to use Qwen-2.5-VL-7B as the judge. When we use the same system prompt as provided to GPT as the judge, the MCQ evaluation result judged by Qwen of edits generated by GPT-Image-1 for example is only ~50%. We suspected it’s because when handling both original image and edited image as inputs, Qwen got distracted. Thus, we conducted a second experiment  by removing the input image, and only making the judge see the edited image. The evaluation results from two runs are 74.0% and 73.5% for edits by GPT-Image-1. This is still largely different from the evaluation result by GPT-4o and Gemini as the judge, which are 85.0% and 82.4% respectively. Upon human checking, we think using Qwen-2.5-vl as the judge is not as accurate.
>
> > Q2: Preservation hinges on the inverted object mask. Small mask errors (under/over-segmentation, ambiguous targets like “sky near horizon”) can mis-score preservation. Quantifying mask quality and its effect (e.g., via perturbation studies) is needed.
>
> A2: We manually checked the masks one by one after generating them, and made sure even for ambiguous terms, the masked area is reasonable. For example, even when asking humans to draw masks on ‘sky near horizon’, humans will return different masks as this task by nature does not lead to ‘one’ precise groundtruth. Thus, it would be hard to quantify how precise each mask is in this case. However, we believe our masks are pushing the image editing evaluation field towards more precise evaluation by providing reasonable enough masks.
>
> > Q3: Single-turn, localized edits only; global style transfer, multi-step, and interactive refinement are out of scope, limiting ecological validity for real editing workflows.
>
> A3: This limitation is inherent to our mask-grounded evaluation design: because the benchmark must specify which region is intended to change, multi-turn or conversational editing is not compatible with pre-defined masks. For multi-turn edits, the target region cannot be known ahead of time and would require dynamically generated masks conditioned on edit history: a fundamentally different benchmark formulation. We will clarify this conceptual distinction. Our goal here is to evaluate atomic, verifiable edits with ground-truth regions, which form the basis for future multi-step evaluation frameworks.
>
> >  Q4: GPT-Image-1 appears in aggregate numbers but is excluded from the human-study pool; this complicates judge↔human comparison for the most capable model.
>
> A4: The human study was conducted early in the project, before GPT-Image-1 became available for external evaluation. We used the originally planned set of models for human annotation to keep the study internally consistent. Since receiving the reviewer’s comment, we have run a small supplementary human study including GPT-Image-1 (100 examples × 3 annotators). We include the result visualization here.
>
> Human vs GPT evaluated functional correctness: https://anonymous.4open.science/r/gie-bench-rebuttal-6F31/human%20vs%20gpt%20evaluated%20functional%20correctness.png
>
> Human evaluated preservation vs inverted MSE rank: https://anonymous.4open.science/r/gie-bench-rebuttal-6F31/human%20evaluated%20preservation%20vs%20inverted%20mse%20rank.png
>
> (to be continued)

---

> > ### Author Response · Authors · 2025-11-22
> > **Response to Reviewer FgRn (Part 2)**
> >
> > >  Q5: Images come from a single stock repository; scene/style diversity and real-world artifact coverage may be narrower than web-scale distributions. Do results hold on other image sources (e.g., LAION subsets, COCO, web photographs) and on global edits (style transfer) when masks are “entire image”?
> >
> > A5: Our primary goal in GIE-Bench was to evaluate edits on high-quality, legally redistributable images, which strongly constrained our choice of source datasets. Public web photographs are often copyrighted, and large web-scale datasets (e.g., LAION) contain a mixture of copyrighted and low-resolution content, making them unsuitable for a benchmark we must fully release. COCO is legally safe but the images are generally lower resolution, noisier, and less suitable for fine-grained editing evaluation. For these reasons, we selected a CC-BY stock repository with consistently high image quality and unambiguous licensing. For global edits such as style transfer where the entire image is the target region, the masked preservation metric naturally collapses to a full-image comparison. Though our images are from a single stock repo, we believe our selected images can still reflect real-world usage well as it covers a diverse range of content.
> >
> > > Q6: Even with 2–5 options, answer distributions or wording may create “easy distractors.” Reporting per-question difficulty and entropy, plus adversarial revisions, would increase robustness.
> >
> > A6: We note that GPT-4o and Gemini do not expose per-option probabilities for MCQs, so we cannot compute entropy directly from logits. GIE-Bench uses MCQs only to verify whether the intended edit occurred, the goal is not to make questions cognitively difficult. For example, if the edit is to add a rabbit on an empty table, the MCQ can be ‘What is on the table?’, and the choices contain ‘rabbit’ and other animals. If the edit was correct, the judge will be able to answer correctly. This method is better than freeform answering because then we will need to compare one or more sentences answered by the judge with the groundtruth answer ‘rabbit’; a step that may cause unnecessary inaccuracy.
> >
> > > Q7:  How sensitive are preservation scores to mask dilation/erosion (±k pixels) and to imperfect detections? Please provide curves and error bars.
> >
> > A7: We show experiment result visualizations here.
> >
> > Sensitivity to mask dilation: https://anonymous.4open.science/r/gie-bench-rebuttal-6F31/sensitivity%20to%20mask%20dilation.png
> >
> > Sensitivity to mask errosion: https://anonymous.4open.science/r/gie-bench-rebuttal-6F31/sensitivity%20to%20mask%20errosion.png
> >
> > The sensitivity curves show that GIE-Bench is stable under realistic mask noise. For dilation (which corresponds to slight over-segmentation), the relative change in preserved-pixel ratio is small across all radii; only 7–15% even at k=5, with consistently low variance. This reflects the fact that most masks are compact, so dilating the boundary by 1–2 pixels affects very few preserved pixels. Erosion behaves as expected: shrinking the mask by 1–2 pixels increases preserved area moderately (14–29%), but larger erosions (k≥3) quickly collapse small masks and therefore produce large relative increases and higher variance. Importantly, the regime corresponding to realistic segmentation errors (1–2 px) shows modest, well-controlled changes, indicating that our preservation metric is robust to typical mask inaccuracies, while the large-k behavior simply reflects extreme, intentionally stressed mask corruption rather than instability in the benchmark.
> >
> > (to be continued)

---

> > > ### Author Response · Authors · 2025-11-22
> > > **Response to Reviewer FgRn (Part 3)**
> > >
> > > > Q9: Which edit types show the largest judge disagreement (GPT-4o vs. Gemini)? Any systematic MCQ failure modes (e.g., spatial terms, size ratios)?
> > >
> > > A9: We present below a table of GPT-4o vs Gemini-Flash Differences (Absolute % Difference)
> > >
> > > | Model             | Add  | Attr | Color | Layout | Replace | Remove | Scene | Size  | Text |
> > > |-------------------|------|------|-------|--------|---------|--------|-------|-------|------|
> > > | StableDiffusion   | 2.50 | 5.00 | 4.17  | 5.00   | 4.17    | 0.83   | 4.17  | 8.33  | 3.33 |
> > > | HQ-Edit           | 7.50 | 3.33 | 8.34  | 0.84   | 3.33    | 3.33   | 3.33  | 10.83 | 3.33 |
> > > | OneDiffusion      | 5.00 | 0.84 | 1.66  | 0.83   | 0.00    | 2.50   | 2.50  | 15.00 | 2.50 |
> > > | InsPix2Pix-CLIP   | 2.50 | 1.67 | 1.66  | 5.00   | 1.66    | 10.00  | 4.17  | 14.16 | 5.00 |
> > > | InsPix2Pix        | 4.17 | 0.00 | 0.84  | 6.67   | 7.50    | 8.34   | 1.67  | 1.67  | 5.83 |
> > > | InsDiffusion      | 8.33 | 3.33 | 10.83 | 7.50   | 5.84    | 5.83   | 6.67  | 9.16  | 0.83 |
> > > | MGIE              | 10.83| 3.33 | 1.67  | 4.17   | 2.50    | 6.67   | 3.34  | 15.00 | 2.50 |
> > > | MagicBrush        | 3.33 | 0.00 | 1.67  | 2.50   | 2.50    | 9.16   | 5.00  | 10.00 | 1.67 |
> > > | OmniGen           | 0.00 | 4.17 | 0.00  | 10.00  | 4.16    | 8.34   | 5.00  | 5.00  | 5.83 |
> > > | GPT-Image-1       | 1.76 | 0.91 | 3.54  | 1.94   | 1.79    | 9.74   | 1.82  | 1.80  | 4.16 |
> > >
> > >
> > > We also present below a table that shows Average Difference per Edit Type Across All Models:
> > >
> > > | Edit Type                   | Avg Δ (%) |
> > > |-----------------------------|-----------|
> > > | Add Object                  | 4.59      |
> > > | Attribute Change            | 2.26      |
> > > | Color Change                | 3.44      |
> > > | Layout Modification         | 4.45      |
> > > | Object Replace              | 3.35      |
> > > | Remove Object               | 6.47      |
> > > | Scene / Background Change   | 3.77      |
> > > | Size Change                 | 9.10      |
> > > | Textual Edit                | 3.50      |
> > >
> > >
> > > Using the average absolute difference across all models:
> > >
> > > Size Change (Δ = 9.10%) : largest cross-judge disagreement
> > >
> > > Remove Object (Δ = 6.47%)
> > >
> > > Add Object (Δ = 4.59%)
> > >
> > > Layout Modification (Δ = 4.45%)
> > >
> > > These categories produce the biggest divergence between GPT-4o and Gemini-Flash judgments, indicating that edits involving geometric transformations, object deletion, and spatial changes are harder to evaluate consistently.
> > >
> > > Regarding systematic MCQ failure: editing systems struggle most with spatial re-layout, precise geometric resizing, and text rendering, regardless of which VLM is used for evaluation.
> > >
> > > > Q10: The calibrated preservation score maps masked MSE to ([0,1]). What normalization is used (per-type, per-image, global), and how does that choice affect rankings?
> > >
> > > A10: Normalization is per-image. Rankings are preserved because every model is evaluated on the same images. Since all models contribute an MSE for each image and the denominator changes only per-image, model-vs-model comparisons remain fair.
> > >
> > >
> > > > Q11: Any preliminary results on chained edits where preservation compounds across steps?
> > >
> > > A11: We did not run multi-step or chained-edit experiments, because the core design of GIE-Bench relies on predefined target masks, which specify exactly which region should change in an edit. In a multi-turn setting, the target region for the second or third edit cannot be known in advance: each step depends on the  result of the previous edit. This makes it impractical to predefine accurate masks for later turns, and thus prevents us from computing grounded preservation metrics in a principled way. For this reason, multi-step evaluation is fundamentally incompatible with the current benchmark formulation, and we intentionally restrict GIE-Bench to single, atomic edits where ground-truth masks can be reliably provided.
> > >
> > > (to be continued)

---

> > > > ### Author Response · Authors · 2025-11-22
> > > > **Response to Reviewer FgRn (Part 4)**
> > > >
> > > > > Q12: Did you analyze category imbalance effects (e.g., human faces vs. landscapes) on preservation/correctness?
> > > >
> > > > A12: Our balancing was done at both the edit-type level and image content level. Below we show the distribution of image content (count of text images is larger than other categories because all images in the text-edit category need to have text content; also text images are present in other edit categories):
> > > >
> > > > | Category            | Count |
> > > > |---------------------|-------|
> > > > | animals             | 51    |
> > > > | jewellery           | 48    |
> > > > | nature              | 52    |
> > > > | musical_instrument  | 57    |
> > > > | plants              | 56    |
> > > > | sign                | 63    |
> > > > | home_appliance      | 48    |
> > > > | art                 | 42    |
> > > > | transportation      | 43    |
> > > > | architecture        | 52    |
> > > > | food                | 51    |
> > > > | accessories         | 41    |
> > > > | human_faces         | 39    |
> > > > | city                | 55    |
> > > > | cartoon             | 34    |
> > > > | human               | 45    |
> > > > | furniture           | 57    |
> > > > | cloth               | 36    |
> > > > | text                | 161   |
> > > > | electronics         | 49    |
> > > > | **All**             | **1080** |
> > > >
> > > >
> > > > We sincerely thank the reviewer for the detailed and constructive suggestions. We will add results of the additional experiments per your request provided above to enhance the paper. We appreciate that the reviewer recognized the strength of GIE-Bench and how it’s advancing the image-editing field.

---

### Official Review · Reviewer_7HhY · 2025-10-30

**Soundness:** 2
**Presentation:** 3
**Contribution:** 2
**Rating:** 2
**Confidence:** 4

**Summary:**

This paper introduces GIE-Bench, a new benchmark for evaluating text-guided image editing models. The authors argue that current evaluation methods, like CLIP scores, are too vague. GIE-Bench gets down to the nitty-gritty with a cool two-pronged approach. First, it checks for "functional correctness" by using an AI-generated multiple-choice question to see if the model actually followed the editing instruction. Second, it measures "content preservation" by using smart object masking to check if the model messed up parts of the image it wasn't supposed to touch. After testing top models like the new GPT-Image-1, they found that while it's great at following orders, it often over-edits the background. GIE-Bench offers a more precise and scalable way to see what these editing models are truly good (and bad) at. However, the paper lacks sufficient innovation.

**Strengths:**

1. It's not just about whether the edit happened, but also about what didn't happen. By separating "functional correctness" from "content preservation," the benchmark gives a much more complete picture of a model's performance.
2.  Using object masks to evaluate only the unedited parts of an image is a brilliant move. It stops penalizing a model for making the correct change and focuses squarely on unintended collateral damage.
3. Fully Automated & Scalable: The entire pipeline—from generating questions and masks to scoring the results—is automated. This makes it easy to use, reproduce, and scale up for testing tons of models on thousands of images.

**Weaknesses:**

1. Only handles one-shot edit. The benchmark is designed for simple, single-step instructions ("change the car to red"). It can't evaluate more complex, real-world scenarios where a user might give a series of commands or have a back-and-forth conversation to refine an image.
2. The introduction of a target mask is of limited importance to the development of current image editing benchmarks.

**Questions:**

1. The paper a key trade-off where models strong at instruction-following (like GPT-Image-1) are weaker at content preservation. Why do you think this is happening? Is it an architectural problem or a fundamental conflict in how these models are trained?
2. Looking ahead, what do you think is the biggest hurdle in expanding GIE-Bench to evaluate multi-turn, conversational image editing? Would it require a completely new way of thinking about evaluation?

---

> ### Author Response · Authors · 2025-11-22
> **Response to Reviewer 7HhY**
>
> We thank the reviewer for the constructive feedback and for recognizing the strengths of GIE-Bench, particularly our separation of functional correctness and content preservation and the benchmark’s fully automated, scalable design. We address the raised concerns below.
>
> > Q1: Only handles one-shot edit. The benchmark is designed for simple, single-step instructions ("change the car to red"). It can't evaluate more complex, real-world scenarios where a user might give a series of commands or have a back-and-forth conversation to refine an image.
>
> A1: We acknowledge that GIE-Bench currently focuses on single-turn edits, and this is an intentional design decision. Because our evaluation relies on predefined object masks, each example must specify exactly which region is intended to change and which should remain intact. In a multi-turn or conversational editing scenario, the subsequent edit cannot be known in advance, making it impossible to predefine accurate masks for all future turns.
> Our goal here is to establish a controlled, ground-truth-verifiable benchmark where correctness and preservation can be objectively measured. GIE-Bench thus serves as a rigorous foundation for evaluating the atomic, single-edit behaviors, which forms the basis for future multi-step evaluation frameworks.
>
> > Q2: The introduction of a target mask is of limited importance to the development of current image editing benchmarks.
>
> A2: We respectfully disagree that introducing target masks is of limited importance. Existing benchmarks (e.g., MagicBrush) evaluate the entire image uniformly, thereby penalizing legitimate edits and rewarding superficial ones. Our object-aware masking directly addresses this long-standing flaw by computing metrics only over unedited regions, yielding a semantically grounded preservation score.
> Empirically, we show that this refinement changes model ranking: GPT-Image-1’s high CLIP score hides substantial over-modification revealed by our masked metric. Thus, the mask is not a minor engineering detail but a key conceptual contribution enabling target-aware evaluation, which we believe is essential for fair benchmarking of editing models.
>
> > Q3: The paper a key trade-off where models strong at instruction-following (like GPT-Image-1) are weaker at content preservation. Why do you think this is happening? Is it an architectural problem or a fundamental conflict in how these models are trained?
>
> A3: We appreciate this insightful question. In principle, a model can excel at both instruction following and content preservation. Because Gpt-Image-1’s image-generation capability is embedded natively inside the 4o LLM, it tends to be very strong at instruction following, but its content preservation performance is comparatively weaker. We believe this comes from model-specific design choices rather than any inherent trade-off.
> As further evidence, models such as Nano-Banana demonstrate that strong instruction adherence and robust content preservation can coexist, reinforcing that the two abilities are not fundamentally at odds.
>
>  > Q4: Looking ahead, what do you think is the biggest hurdle in expanding GIE-Bench to evaluate multi-turn, conversational image editing? Would it require a completely new way of thinking about evaluation?
>
> A4: Scaling GIE-Bench to multi-turn editing indeed poses new challenges around state tracking and referential grounding (e.g., “now make that object brighter”). The masks provided in GIE-Bench are predefined; they are first computed by DINO+SAM, then filtered by human for quality. In a multi-turn scenario, masks need to be automatically computed after each turn is edited based on the specific edited image. The challenge lies in finding a way to automatically and precisely compute the mask for area that needs to be preserved.
>
> We thank the reviewer again for the detailed and balanced assessment. We hope this clarification shows that GIE-Bench makes a substantive and forward-looking contribution toward rigorous, grounded evaluation of text-guided image editing.

---

### Meta-Review · Area_Chair_kneZ · 2025-12-30

**Summary:**

The paper initially received two negative and one positive ratings. The concerns are mostly about 1) limitation to only handle the single-turn, one-shot edit, 2) judgment protocol beyond GPT/Gemini, 3) sensitivity to mask, 4) more analysis, e.g., types of edit, 5) consistency with human evaluation.

**Reviewer Concerns:**

The authors have provided responses in the rebuttal to answer initial concerns from the reviewers. The AC took a close look at the paper, reviews, and the rebuttal. After the rebuttal, the AC finds that some questions are addressed with more experiments, e.g., judgment protocol, sensitivity to mask, and more analysis. However, as all the reviewers pointed out, the limitation of only handling the single-turn, one-shot edit is significant, in which the authors have not provided a reasonable solution on top of the proposed benchmarking way. Moreover, only having 100 examples to evaluate whether the benchmark is aligned with human evaluation is limited, which is an essential task for any new benchmark. Considering these two major concerns not being addressed well in the rebuttal, the AC agrees with the reviewers' overall feedback and hence recommends the rejection rating.

**Reviewer Scores:**

Reviewer CQfu mentioned to retain the original rating, while the other two reviewers did not fully participate the discussion.

---

### Decision · Program_Chairs · 2026-01-26

Reject